# Dialog without Dialog Data: Learning Visual Dialog Agents from VQA Data

**Michael Cogswell**[5*] **Jiasen Lu**[3*] **Rishabh Jain**[1] **Stefan Lee**[2] **Dhruv Batra**[4,1] **Devi Parikh**[4,1]

[1]Georgia Institute of Technology    [2]Oregon State University
[3] Allen Institute for AI    [4] Facebook AI Research    [5] SRI International
michael.cogswell@sri.com jiasenl@allenai.org leestef@oregonstate.edu
{rishabhjain, dbatra, parikh}@gatech.edu

## Abstract

Can we develop visually grounded dialog agents that can efficiently adapt to new tasks without forgetting how to talk to people? Such agents could leverage a larger variety of existing data to generalize to new task, minimizing expensive data collection and annotation. In this work, we study a setting we call "*Dialog without Dialog*", which requires agents to develop visually grounded dialog models that can adapt to new tasks without language level supervision. By factorizing intention and language, our model minimizes linguistic drift after fine-tuning for new tasks. We present qualitative results, automated metrics, and human studies that all show our model can adapt to new tasks and maintain language quality. Baselines either fail to perform well at new tasks or experience language drift, becoming unintelligible to humans. Code has been made available at: `https://github.com/mcogswell/dialog_without_dialog`.

## 1 Introduction

One goal of AI is to enable humans and computers to communicate naturally with each other in grounded language to achieve a collaborative objective. Recently the community has studied goal oriented dialog, where agents communicate for tasks like booking a flight or searching for images [1].

A popular approach to these tasks has been to observe humans engaging in dialogs like the ones we would like to automate and then train agents to mimic these human dialogs [2, 3]. Mimicking human dialogs allows agents to generate intelligible language (*i.e.*, meaningful English, not gibberish). However, these models are typically fragile and generalize poorly to new tasks. As such, each new task requires collecting new human dialogs, which is a laborious and costly process often requiring many iterations before high quality dialogs are elicited [4, 5].

A promising alternative is to use goal completion as a supervisory signal to adapt agents to new tasks. Specifically, this is realized by pre-training dialog agents via human dialog supervision on one task and then fine-tuning them on a new task by rewarding the agents for solving the task regardless of the dialog's content. This approach can indeed improve task performance, but language quality suffers even for similar tasks. It tends to drifts from human language, becoming ungrammatical and loosing human intelligible semantics – sometimes even turning into unintelligible code. Such code may allow communication with other bots, but is largely incomprehensible to humans. This trade off between task performance and language drift has been observed in prior dialog work [2, 3].

The goal of this paper is to develop visually grounded dialog models that can adapt to new tasks while exhibiting less linguistic drift, thereby reducing the need to collect new data for the new tasks. To test this we consider an image guessing game demonstrated in Fig. 1 (right). In each episode, one agent (A-bot in red) secretly selects a target image $y$ (starred) from a pool of images. The other agent

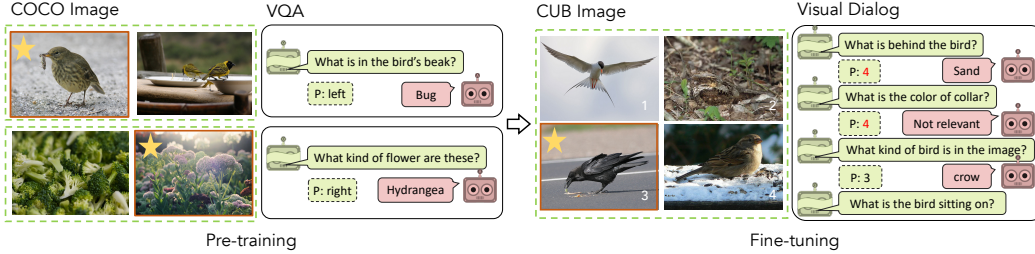

Pre-training                                          Fine-tuning

Figure 1: An example of *Dialog without Dialog* (DwD) task: (Left) We pre-train questioner agent (Q-bot in green) that can discriminate between pairs of images by mimicking questions from VQAv2 [6]. (Right) Q-bot needs to generate a sequence of discriminative questions (a dialog) to identify the secret image that A-bot picked. Note that the language supervision is not available, thus we can only fine-tune Q-bot with task performance. In DwD, we can test Q-bot generalization ability by varying dialog length, pool size and image domain.

(Q-bot in green) must identify this image by asking questions for A-bot to answer in a dialog. To succeed, Q-bot needs to understand the image pool, generate discriminative questions, and interpret the answers A-bot provides to identify the secret image. The image guessing game provides the agent with a goal, and we can test Q-bot generalization ability by varying dialog length, pool size and image domain.

**Contribution 1.** We propose the *Dialog without Dialog* (DwD) task, which requires a Q-bot to perform our image guessing game without dialog level language supervision. As shown in in Fig. 1 (left), Q-bot in this setting first learns to ask questions to identify the secret image by mimicking single-round human-annotated visual questions. For the dialog task (right), no human dialogs are available so Q-bot can only be supervised by its image guessing performance. To measure task performance and language drift in increasingly out-of-distribution settings we consider varied pool sizes and take pool images from diverse image sources (*e.g.* close-up bird images).

**Contribution 2.** We propose a Q-bot architecture for the DwD task that decomposes question intent from the words used to express that intent. We model the question intent by introducing a discrete latent representation that is the only input to the language decoder. We further pair this with a *pre-train then fine-tune* learning approach that teaches Q-bot how to ask visual questions from VQA during pre-training and 'what to ask' during fine-tuning for visual dialogs.

**Contribution 3.** We measure Q-bot's ability to adapt to new tasks and maintain language quality. Task performance is measured with both automatic and human answerers while language quality is measured using three automated metrics and two human judgement based metrics. Our results show the proposed Q-bot both adapts to new tasks better than a baseline chosen for language quality and maintains language quality better than a baseline optimized for just task performance.

## 2 Dialog Based Image Guessing Game

### 2.1 Game Definition

Our image guessing game proceeds one round at a time, starting at round $r = 1$ and running for a fixed number of rounds of dialog $R$. At round $r$, Q-bot observes the pool of images $\mathcal{I} = \{I_1, \ldots, I_P\}$ of size $P$, the history of question answer pairs $q_1, a_1, \ldots q_{r-1}, a_{r-1}$, and placeholder representations $q_0, a_0$ that provide input for the first round. It generates a question

$$q_r = \texttt{QBot.Ask}(\mathcal{I}, q_0, a_0, \ldots q_{r-1}, a_{r-1}). \tag{1}$$

Given this question, but not the entire dialog history, A-bot answers based on the randomly selected target image $I_{\hat{y}}$ (not known to Q-bot):

$$a_r = \texttt{ABot.Answer}(I_{\hat{y}}, q_r). \tag{2}$$

Once Q-bot receives the answer from A-bot, it makes a prediction $y_r$ guessing the target image:

$$y_r = \texttt{QBot.Predict}(\mathcal{I}, q_0, a_0, \ldots, q_r, a_r) \tag{3}$$

**Comparison to GuessWhich.** Our Image Guessing game is inspired by GuessWhich game of Das et al. [2], and there are two subtle but important differences. In GuessWhich, Q-bot initially observes

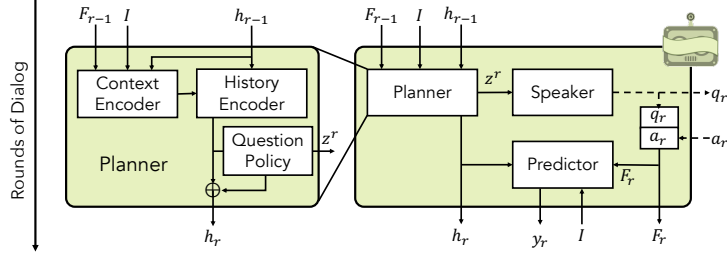

Figure 2: A single round of our Q-bot which decomposes into the modules described in Section 2.3. This factorization allows us to limit language drift while fine-tuning for task performance.

a caption describing A-bot's selected image and must predict the selected image's features to retrieve it from a large, fixed pool of images it does not observe. First, the inclusion of the caption leaves little room for the dialog to add information [7], so we omit it. Second, in our game a small pool of images is sampled for each dialog and Q-bot directly predicts the target image given those choices.

## 2.2 Modelling A-bot

In this work, we focus primarily on Q-bot agent rather than A-bot. We set A-bot to be a standard visual question answering agent, specifically the Bottom-up Top-down [8] model; however, we do make one modification. Q-bot may generate questions that are not well grounded in A-bot's selected image (though they may be grounded in other pool images) – e.g. asking about a surfer when none exists. To enable A-bot to respond appropriately, we augment A-bot's answer space with a `Not Relevant` token. To generate training data for this token we augment every image with an additional randomly sampled question and set `Not Relevant` as its target answer. A-bot is trained independently from Q-bot on the VQAv2 dataset and then frozen.

## 2.3 Modelling Q-bot

We conceptualize Q-bot as having three major modules. The *planner* encodes the state of the game to decide what to ask about. The *speaker* takes this intent and formulates the language to express it. The *predictor* makes target image predictions taking the dialog history into account. We make fairly standard design choices here, then adapt this model for the DwD task in Section 3.

**Pool & Image Encoding.** We represent the $p$-th image $I_p$ of the pool as a set of $B$ bounding boxes such that $I_b^p$ is the embedding of the $b$-th box using the same Faster R-CNN [9] embeddings as in [10]. Note that we do not assume prior knowledge about the size or composition of the pool.

**Planner.** The planner's role is to encode the dialog context (image pool and dialog history) into representation $z^r$, deciding what to ask about in each round. It also produces an encoding $h_r$ of the dialog history. To limit clutter, we denote the question-answer pair at round $r$ as a 'fact' $F_r = [q_r, a_r]$.

**Planner – Context Encoder.** Given the prior dialog state $h_{r-1}$, $F_{r-1}$, and image pool $\mathcal{I}$, the context encoder performs hierarchical attention over images in $\mathcal{I}$ and object boxes in each image to identify image regions that are most relevant for generating the next question. As we detail in Section 2 of the supplement, $F_{r-1}$ and $h_{r-1}$ query the image to attend to relevant regions across the pool. First a $P$ dimensional distribution $\alpha$ over images in the pool is produced and then a $B$ dimensional distribution $\beta^p$ over boxes is produced for each image $p \in \{1, \ldots, P\}$. The image pool encoding $\hat{v}_r$ at round $r$ is

$$\hat{v}_r = \sum_{p=1}^{P} \alpha_p \sum_{b=1}^{B} \beta_b^p I_b^p. \tag{4}$$

This combines the levels of attention and is agnostic to pool size.

**Planner – History Encoder.** To track the state of the game, the planner applies an LSTM-based history encoder that takes $\hat{v}_r$ and $F_{r-1}$ as input and produces an intermediate hidden state $h_r$. Here $h_r$ includes a compact representation of question intent and dialog history, helping provide a differentiable connection between the intent and final predictions through the dialog state.

**Planner – Question Policy.** The question policy transforms $h_r$ to this module's output $z^r$, which the speaker decodes into a question. By default $z^r$ is equal to the hidden state $h_r$, but in Section 3.2 we show how a discrete representations can be used to reduce language drift.

**Speaker.** Given an intent vector $z^r$, the speaker generates a natural language question. Our speaker is a standard LSTM-based decoder with an initial hidden state equal to $z^r$.

**Predictor.** The predictor uses the dialog context generated so far to guess the target image. It takes a concatenation $F = [F_1, \ldots, F_r]$ of fact embeddings and the dialog state $h_r$ and computes an attention pooled fact $\hat{F}$ using $h_r$ as attention context. Along with $h_r$, this is used to attend to salient image features then compute a distribution over images in the pool using a softmax (see Algorithm 2 in the supplement for full details), allowing for the use of cross-entropy as the task loss. Note that the whole model is agnostic to pool size.

## 3 Dialog without Dialog

Aside from some abstracted details, the game setting and model presented in the previous section could be trained without any further information – a pool of images could be generated, A-bot could be assigned an image, the game could be rolled out for arbitrarily many rounds, and Q-bot could be trained to predict the correct image given A-bot's answers. While this is an interesting research direction in its own right [11, 12, 13], there is an obvious shortcoming – it would be highly improbable for Q-bot to discover a fully functional language that humans can already understand. Nobody discovers French. They have to learn it.

At the other extreme – representing standard practice in dialog problems – humans could be recruited to perform this image guessing game and provide dense supervision for what questions Q-bot should ask to perform well at this specific task. However, this requires collecting language data for every new task. It is also intellectually dissatisfying for agents' knowledge of natural language to be so inseparably intertwined with individual tasks. After all, one of the greatest powers of language is the ability to use it to communicate about many different problems.

In this section, we consider a middle ground that has two stages. Stage 1 trains our agent on one task where training data already exists (VQA; *i.e.*, single round dialog) and then stage 2 adapts it to carry out goal driven dialog (image guessing game) without further supervision.

### 3.1 Stage 1: Language Pre-training

We leverage the VQAv2 [6] dataset as our language source to learn how to ask questions that humans can understand. By construction, for each question in VQAv2 there exists at least one pair of images which are visually similar but have different ground truth answers to the same question. Fortuitously, this resembles our dialog game – the image pair is the pool, the question is guaranteed to be discriminative, and we can provide an answer depending on A-bot's selected image. We view this as a special case of our game that is fully supervised but contains only a single round of dialog. During stage 1 Q-bot is trained to mimic the human question (via cross-entropy teacher forcing) and to predict the correct image given the ground truth answer.

For example, in the top left of Fig. 1 outlined in dashed green we show a pair of two bird images with the question "What is in the bird's beak?" from VQAv2. Our agents engage in a single round dialog where Q-bot asks that question and A-bot provides the answer (also supervised by VQAv2).

### 3.2 Stage 2: Transferring to Dialog

A first approach for adapting agents would be to take the pre-trained weights from stage 1 and simply fine-tune for our full image guessing task. However, this agent would face a number of challenges. It has never had to model multiple steps of a dialog. Further, while trying to predict the target image there is little to encourage Q-bot to continue producing intelligible language. Indeed, we find our baselines do exhibit language drift. We consider four modifications to address these problems.

**Discrete Latent Intention Representation** $z^r$. Rather than a continuous vector passing from the question policy to the speaker, we pass discrete vectors. Specifically, we consider a representation composed of $N$ different $K$-way Concrete variables [14]. Let $z_n^r \in \{1, \ldots, K\}$ and let the logits $l_{n,1}, \ldots, l_{n,K}$ paramterize the Concrete distribution $p(z_n^r)$. We learn a linear transformation $W_n^z$ from the intermediate dialog state $h_r$ to produce these logits for each variable $n$:

$$l_{n,k} = \text{LogSoftmax} \left( W_n^z h_r \right)_k \ \forall k \in \{1, \ldots, K\} \ \forall n \in \{1, \ldots, N\} \tag{5}$$

To provide input to the speaker, $z^r$ is embedded using a learned dictionary of embeddings. In our case each variable in $z^r$ has a dictionary of $K$ learned embeddings. The value of $z_n^r \ (\in \{1, \ldots, K\})$ picks

one of the embeddings for each variable and the final representation simply sums over all variables:

$$e_z = \sum_{n=1}^{N} E_n(z_n^r).$$ (6)

**VAE Pre-training.** When using this representation for the intent, we train stage 1 by replacing the likelihood with an ELBO (Evidence Lower BOund) loss as seen in Variational Auto-Encoders (VAEs) [15] to help disentangle intent from expression by restricting information flow through $z^r$. We use the existing speaker module to decode $z^r$ into questions and train a new encoder module to encode ground truth VQAv2 question $\hat{q}_1$ into conditional distribution $q(z^1|\hat{q}_1, \mathcal{I})$ over $z^r$ at round 1.

For the encoder we use a version of the previously described context encoder from Section 2.3 that uses the question $\hat{q}_1$ as attention query instead of $F_{r-1}$ and $h_{r-1}$ (which are not available in this context). The resulting ELBO loss is

$$\mathcal{L} = E_{z^1 \sim q(z^1|\hat{q}_1, \mathcal{I})} \left[\log p(\texttt{speaker}(z^1))\right] + \frac{1}{N} \sum_{n=1}^{N} D_{KL} \left[q(z_n^1|\hat{q}_1, \mathcal{I})||\mathcal{U}(K)\right]$$ (7)

This is like the Full ELBO described, but not implemented, in [16]. The first term encourages the encoder to represent and the speaker to mimic the VQA question. The second term uses the KL Divergence $D_{KL}$ to push the distribution of $z$ close to the $K$-way uniform prior $\mathcal{U}(K)$, encouraging $z$ to ignore irrelevant information. Together, the first two terms form an ELBO on the question likelihood given the image pool [17, 18].

**Fixed Speaker.** Since the speaker contains only lower level information about how to generate language, we freeze it during task transfer. We want only the high level ideas represented by $z$ and the predictor which receives direct feedback to adapt to the new task. If we updated the speaker then its language could drift given only the sparse feedback available in each new setting.

**Adaptation Curriculum.** As the pre-trained (stage 1) model has never had to keep track of dialog contexts beyond the first round, we fine-tune in two stages, 2.A and 2.B. In **stage 2.A** we fix the Context Encoder and Question Policy parts of the planner so the model can learn to track dialog effectively without trying to generate better dialog at the same time. This stage takes 20 epochs to train. Once Q-bot learns how to track dialog we update the entire planner in **stage 2.B** for 5 epochs.[2]

# 4 Experiments

We want to show that our proposed agent can adapt to new tasks while exhibiting less linguistic drift. In Section 4 and Section 4.1 we start by describing the new tasks we construct and the baselines we compare to, then the following sections demonstrate how our model adapts while preventing drift using qualitative examples (Section 4.2), automated metrics (Section 4.3), and human judgements (Section 4.4). We also summarize the model ablations (Section 4.5) detailed in the supplement.

**Task Settings.** We construct new tasks by varying four parameters of our image guessing game:

- **Number of Dialog Rounds.** The number of dialog rounds $R$ is fixed at 1, 5, or 9.
- **Pool Size.** The number of images in a pool $P$ to 2, 4, or 9.
- **Image Domain.** By default we use VQA images (*i.e.*, from COCO [19]), but we also construct pools using CUB (bird) images [20] and AWA (animal) images [21].
- **Pool Sampling Strategy.** We test two ways of sampling pools of images. The Constrast sampling method, required for pre-training (Section 3.1), chooses a pair of images with contrasting answers to the same question from VQAv2. This method only works for $P = 2$. The Random sampling method chooses $P$ images at random from the images available in the split.

For example, consider the 'VQA - 2 Contrast - 5 Round' setting. These pools are constructed from 2 VQA images with the Contrast sampling strategy and dialogs are rolled out to 5 rounds.

## 4.1 Baselines

We compare our proposed approach to two baselines – **Zero-shot Transfer** and **Typical Transfer** – ablating aspects of our model that promote adaptation to new tasks or prevent language drift.

The **Zero-shot Transfer** baseline is our model after the single round fully supervised pre-training. Improvements over this model represent gains made from task based fine-tuning. The **Typical Transfer** baseline is our model under standard encoder-decoder dialog model design choices – *i.e.*, a continuous latent variable, maximum likelihood pre-training, and fine-tuning the speaker module. Improvements over this model represent gains made from the modifications aimed at preventing language drift described in Section 3.2 – specifically, the discrete latent variable, ELBO pre-training, and frozen speaker module.

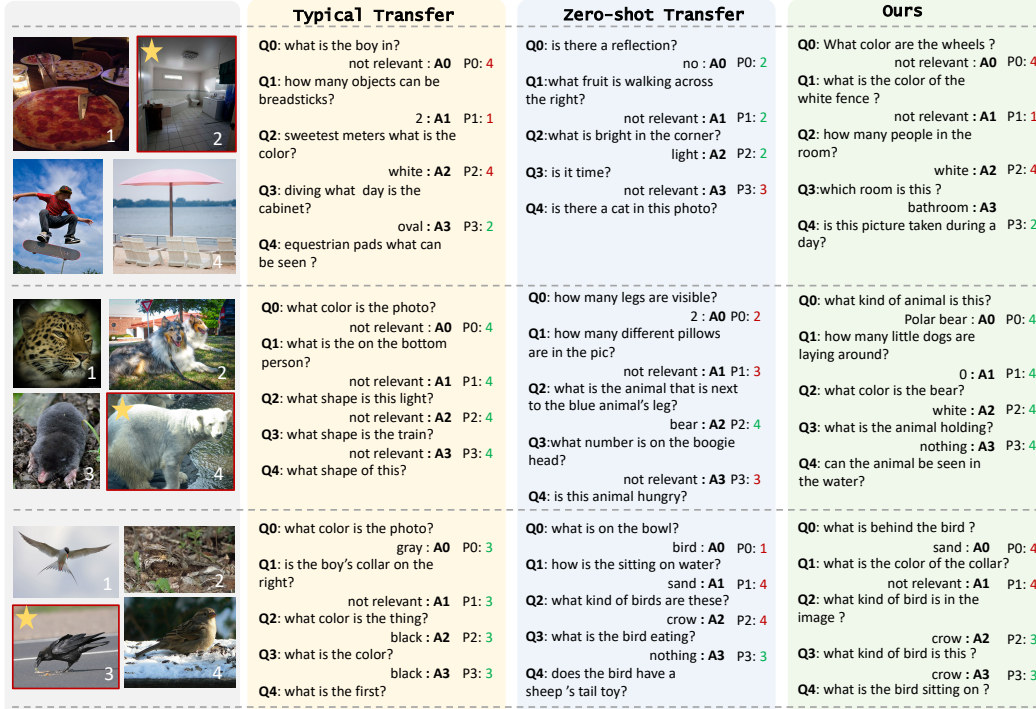

| Typical Transfer | Zero-shot Transfer | Ours |
|---|---|---|
| **Q0**: what is the boy in?<br>not relevant : **A0**  P0: 4<br>**Q1**: how many objects can be breadsticks?<br>2 : **A1**  P1: 1<br>**Q2**: sweetest meters what is the color?<br>white : **A2**  P2: 4<br>**Q3**: diving what  day is the cabinet?<br>oval : **A3**  P3: 2<br>**Q4**: equestrian pads what can be seen ? | **Q0**: is there a reflection?<br>no : **A0**  P0: 2<br>**Q1**:what fruit is walking across the right?<br>not relevant : **A1**  P1: 2<br>**Q2**:what is bright in the corner?<br>light : **A2**  P2: 2<br>**Q3**: is it time?<br>not relevant : **A3**  P3: 3<br>**Q4**: is there a cat in this photo? | **Q0**: What color are the wheels ?<br>not relevant : **A0**  P0: 4<br>**Q1**: what is the color of the white fence ?<br>not relevant : **A1**  P1: 1<br>**Q2**: how many people in the room?<br>white : **A2**  P2: 4<br>**Q3**:which room is this ?<br>bathroom : **A3**<br>**Q4**: is this picture taken during a  P3: 2<br>day? |
| **Q0**: what color is the photo?<br>not relevant : **A0**  P0: 4<br>**Q1**: what is the on the bottom person?<br>not relevant : **A1**  P1: 4<br>**Q2**: what shape is this light?<br>not relevant : **A2**  P2: 4<br>**Q3**: what shape is the train?<br>not relevant : **A3**  P3: 4<br>**Q4**: what shape of this? | **Q0**: how many legs are visible?<br>2 : **A0**  P0: 2<br>**Q1**: how many different pillows are in the pic?<br>not relevant : **A1**  P1: 3<br>**Q2**: what is the animal that is next to the blue animal's leg?<br>bear : **A2**  P2: 4<br>**Q3**:what number is on the boogie head?<br>not relevant : **A3**  P3: 3<br>**Q4**: is this animal hungry? | **Q0**: what kind of animal is this?<br>Polar bear : **A0**  P0: 4<br>**Q1**: how many little dogs are laying around?<br>0 : **A1**  P1: 4<br>**Q2**: what color is the bear?<br>white : **A2**  P2: 4<br>**Q3**: what is the animal holding?<br>nothing : **A3**  P3: 4<br>**Q4**: can the animal be seen in the water? |
| **Q0**: what color is the photo?<br>gray : **A0**  P0: 3<br>**Q1**: is the boy's collar on the right?<br>not relevant : **A1**  P1: 3<br>**Q2**: what color is the thing?<br>black : **A2**  P2: 3<br>**Q3**: what is the color?<br>black : **A3**  P3: 3<br>**Q4**: what is the first? | **Q0**: what is on the bowl?<br>bird : **A0**  P0: 1<br>**Q1**: how is the sitting on water?<br>sand : **A1**  P1: 4<br>**Q2**: what kind of birds are these?<br>crow : **A2**  P2: 4<br>**Q3**: what is the bird eating?<br>nothing : **A3**  P3: 3<br>**Q4**: does the bird have a sheep 's tail toy? | **Q0**: what is behind the bird ?<br>sand : **A0**  P0: 4<br>**Q1**: what is the color of the collar?<br>not relevant : **A1**  P1: 4<br>**Q2**: what kind of bird is in the image ?<br>crow : **A2**  P2: 3<br>**Q3**: what kind of bird is this ?<br>crow : **A3**  P3: 3<br>**Q4**: what is the bird sitting on ? |

Figure 3: Qualitative comparison of dialogs generated by our model with those generated by Typical Transfer and Zero-shot Transfer baselines. Top / middle / bottom rows are image pool from COCO / AWA / CUB images respectively. Our model is pre-trained on VQA (COCO images) and generates more intelligible questions on out-of-domain images.

## 4.2   Qualitative Results

Figure 3 shows example outputs of the Typical Transfer and Zero-shot Transfer baselines alongside our Q-bot on VQA, AWA and CUB images using size 4 Randomly sampled pools and 5 rounds of dialog. Both our model and the Typical Transfer baseline tend to guess the target image correctly, but it is much easier to tell what the questions our model asks mean and how they might help with guessing the target image. On the other hand, questions from the Zero-shot Transfer baseline are clearly grounded in the images, but they do not seem to help guess the target image and the Zero-shot Transfer baseline indeed fails to guess correctly. This is a pattern we will reinforce with quantitative results in Section 4.4 and Section 4.3.

These examples and others we have observed suggest interesting patterns that highlight A-bot. Our automated A-bot based on [10] does not always provide accurate answers, limiting the questions Q-bot can usefully ask. When there is signal in the answers, it is not necessarily intelligible, providing an opportunity for Q-bot's language to drift.

## 4.3   Automated Evaluation

We consider metrics addressing both **Task** performance and **Language** quality. While task performance is straightforward (did Q-bot guess the correct target image?), language quality is harder to measure. We describe three automated metrics here and further investigate language quality using human evaluations in Section 4.4.

**Task – Guessing Game Accuracy.** To measure task performance so we report the accuracy of Q-bot's target image guess at the final round of dialog.

**Language – Question Relevance via A-bot.** To be human understandable, the generated questions should be relevant to at least one image in the pool. We measure question relevance as the maximum question image relevance across the pool as measured by A-bot, i.e., $1 - p(\texttt{Not Relevant})$. We note that this is only a proxy for actual question relevance as A-bot may report `Not Relevant` erroneously if it fails to understand Q-bot's question; however, in practice we find A-bot does a fair job in determining relevance. We also provide human relevance judgements in Section 4.4.

**Language – Fluency via Perplexity.** To evaluate Q-bot's fluency, we train an LSTM-based language model on the corpus of questions in VQA. This allows us to evaluate the perplexity of the questions generated by Q-bot for dialogs on its new tasks. Lower perplexity indicates the generated questions are similar to VQA questions in terms of syntax and content. However, we note that questions generated for the new tasks could have lower perplexity because they have drifted from English or because different things must be asked for the new task, so lower perplexity is not always better [22].

**Language – Diversity via Distinct $n$-grams.** This considers the set of all questions generated by Q-bot across all rounds of dialog on the val set. It counts the number of $n$-grams in this set, $G_n$, and the number of distinct $n$-grams in this set, $D_n$, then reports $\frac{G_n}{D_n}$ for each value of $n \in \{1, 2, 3, 4\}$. Note that instead of normalizing by the number of words as in previous work [23, 24], we normalize by the number of n-grams so that the metric represents a percentage for values of $n$ other than $n = 1$. Generative language models frequently produce safe standard outputs [23], so diversity is a sign this problem is decreasing, but diversity by itself does not make language meaningful or useful.

**Results.** Table 1 presents results on our val set for our model and baselines across the various settings described in Section 4. Agents are tasked with generalizing further and further from their source language data. Setting A is the same as for stage 1 pre-training. In that same column, B and C require generalization to multiple rounds of dialog and Randomly sampled image pairs instead of pools sampled with the Contrast strategy. In the right side of Tab. 1 we continue to test generalization farther from the language source using more images and rounds of dialog (D) and then using different types of images (E and F). Our model performs well on both task performance and language quality across the different settings in terms of these automatic evaluation metrics. Other notable findings are:

| | | | Accuracy ↑ | Perplexity↓ | Relevance ↑ | Diversity↑ | | | | Accuracy ↑ | Perplexity↓ | Relevance ↑ | Diversity↑ |
|---|---|---|---|---|---|---|---|---|---|---|---|---|---|
| VQA 2 Contrast 1 Round | A1 | Zero-shot Transfer | 0.73 | 2.62 | 0.87 | 0.50 | VQA 9 Random 9 Rounds | D1 | Zero-shot Transfer | 0.18 | 2.72 | **0.77** | 1.11 |
| | A2 | Typical Transfer | 0.71 | 10.62 | 0.66 | 5.55 | | D2 | Typical Transfer | **0.78** | 40.66 | **0.77** | 2.57 |
| | A3 | Ours | **0.82** | **2.6** | **0.88** | 0.54 | | D3 | Ours | 0.53 | **2.55** | 0.75 | 0.95 |
| VQA 2 Contrast 5 Rounds | B1 | Zero-shot Transfer | 0.67 | 2.62 | 0.87 | 0.50 | AWA 9 Random 9 Rounds | E1 | Zero-shot Transfer | 0.47 | 2.49 | **0.96** | 0.24 |
| | B2 | Typical Transfer | 0.74 | 10.62 | 0.66 | 5.55 | | E2 | Typical Transfer | 0.48 | 12.56 | 0.64 | 2.21 |
| | B3 | Ours | **0.87** | 2.60 | **0.88** | 0.54 | | E3 | Ours | **0.74** | 2.41 | **0.96** | 0.28 |
| VQA 2 Random 5 Rounds | C1 | Zero-shot Transfer | 0.64 | **2.64** | 0.75 | 1.73 | CUB 9 Random 9 Rounds | F1 | Zero-shot Transfer | 0.36 | 2.56 | **1.00** | 0.04 |
| | C2 | Typical Transfer | 0.86 | 16.95 | 0.62 | 8.13 | | F2 | Typical Transfer | 0.38 | 20.92 | 0.47 | 2.16 |
| | C3 | Ours | **0.95** | 2.69 | **0.77** | 2.34 | | F3 | Ours | **0.74** | **2.47** | **1.00** | 0.04 |

Table 1: Performance of our models and baselines in different experimental settings. From setting A to setting F, agents are tasked with generalizing further from the source data. Our method strikes a balance between guessing game performance and interpretability.

**Ours *vs*. Zero-shot Transfer.** To understand the relative importance of the proposed stage 2 training which transferring to dialog for DwD task, we compared the task accuracy of our model with that of Zero-shot Transfer. In setting, A which matches the training regime, our model outperforms Zero-shot Transfer by 9% (A3 *vs*. A1) on task performance. As the tasks differ in settings B-F, we see further gains with our model consistently outperforming Zero-shot Transfer by 20-38%. Despite these gains, our model maintains similar language perplexity, A-bot relevance, and diversity.

**Ours *vs*. Typical Transfer.** Our discrete latent variable, variational pre-training objective, and fixed speaker play an important role in avoiding language drift. Compared to the Typical Transfer model without these techniques, our model achieves over 4x (A2 / A3) lower perplexity and 10-53% better A-bot Relevance. Our model also improves in averaged accuracy, which means more interpretable language also improves the task performance. Note that Typical Transfer has 2-100x higher diversity compared to our model, which is consistent with the gibberish we observe from that model (*e.g.*, in Fig. 3) and further suggests its language is drifting away from English.

**Results from Game Variations.** We consider the following variations on the game:

– **Dialog Rounds.** Longer dialogs (more rounds) achieve better accuracy (A3 vs B3).

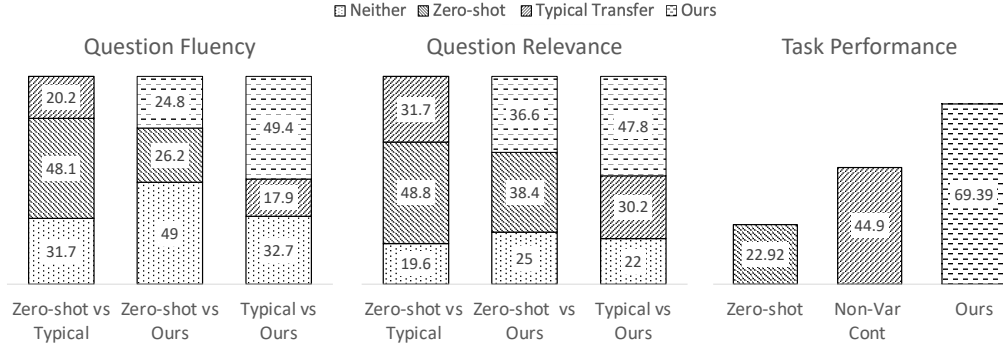

Figure 4: Human evaluation of language quality – question fluency (left), relevance (middle) and task performance (right). Question fluency and relevance compare a pair of agent-generated questions, asking users which (or possibly neither) is more fluent/relevant. Task performance is to have humans interact dynamically with Q-bot in real time.

– **Pool Sampling Strategy.** As expected, Random pools are easier compared to Contrast pools (B3 vs C3 accuracy), however language fluency and relevance drop on the Random pools (B3 vs C3 perplexity and a-bot relevance).

– **Image Source.** CUB and AWA pools are harder compared to COCO image domain (D3 vs E3 vs F3). Surprisingly, our models maintains similar perplexity and high a-bot relevance even on these out-of-domain image pools. The Zero-shot Transfer and Typical Transfer baselines generalize poorly to these different image domains – reporting task accuracies nearly half our model performance.

## 4.4 Human Studies

**Human Study for Question Relevance.** To get a more accurate measure of question relevance, we asked humans to evaluate questions generated by our model and the baselines (Zero-shot Transfer & Typical Transfer). We curated 300 random, size 4 pools where all three models predicted the target correctly at round 5. For a random round, we show turkers the questions from a pair of models and ask *'Which question is most relevant to the images?'* Answering the question is a forced choice between three options: one of the two models or an *'Equally relevant'* option. The results in Fig. 4 (middle) show the frequency with which each option was chosen for each model pair. Our model was considered more relevant than the Typical Transfer model (47.8% *vs*. 30.2%) and about the same as the Zero-shot Transfer model (36.6% *vs*. 38.4%).

**Human Study for Question Fluency.** We also evaluate fluency by asking humans to compare questions. In particular, we presented the same pairs of questions to turkers as in the relevance study, but this time we did *not* present the pool of images and asked them *'Which question is more understandable?'* As before, there was a forced choice between two models and an *'Equally understandable'* option. This captures fluency because humans are more likely to report that they understand grammatical and fluent language. We used the same pairs of questions as in the relevance interface but turkers were not given image pools with which to associate the questions. As in the relevance study, questions were presented in a random order.

Figure 4 (left) shows the frequency with which each option was chosen for each model pair. Our model is considered more fluent than the Typical Transfer model (49.4% *vs*. 17.9%) and about the same as the Zero-shot Transfer model (24.8% *vs*. 26.2%).

**Human Study for Task Performance.** What we ultimately want in the long term is for humans to be able to collaborate with bots to solve tasks. Therefore, the most direct evaluation of our the DwD task is to have humans interact dynamically with Q-bot. We implemented an interface that allowed turkers to interact with Q-bot in real time, depicted in Fig. **??**. Q-bot asks a question. A human answers it after looking at the target image. Q-bot asks a new question in response to the human answer and the human responds to that question. After the 4th answer Q-bot makes a guess about which target image the human was answering based on.

We perform this study for the same pools for each model and find our approach achieves an accuracy of 69.39% – significantly higher than Typical Transfer at 44.90% and Zero-shot Transfer at 22.92% as shown in Fig. 4 (right). This study shows that our model learns language for this task that is amenable to human-AI collaboration. This is in contrast to prior work [25] that showed that improvements captured by task-trained models for similar image-retrieval tasks did not transfer when paired with human partners.

## 4.5 Model Ablations

We investigate the impact of our modelling choices from Section 3.2 by ablating these choices in Section 5 of the supplement, summarizing the results here. The choice of discrete instead of continuous $z^r$ helps maintain language quality, as does the use of variational (ELBO) pre-training instead of maximum likelihood. Surprisingly, the ELBO loss probably has more impact than the discreteness of $z^r$. Fixing the speaker module during stage 2 also had a minor role in discouraging language drift. Finally, we find that improvements in task performance are due more to learning to track the dialog in stage 2.A than they are due to asking more discriminative questions.

## 5 Related Work

Our interest comes from language drift problems encountered when using models comparable to the Zero-shot Transfer baseline. In [3] a dataset is collected with question supervision then fine-tuning is used in an attempt to increase task performance, but the resulting utterances are unintelligible. Similarly, [2] takes a very careful approach to fine-tuning for task performance but finds that language also diverges, becoming difficult for humans to understand. Neither approach uses a discrete latent variables or a multi stage training curriculum, as in our proposed model. Furthermore, these models need to be adapted to work in our new setting, and doing so would yield models very similar to our Typical Transfer baseline.

More recently, [26] observe language drift in a translation game from French to German. They reduce drift by supervising communications between agents with auxiliary translations to English and grounding in images. This setting is somewhat different than ours since grounding is directly necessary to solve our task. The approach also requires direct supervision on the communication channel, which is not practical for a multiple round dialog game like ours.

We used a visual reference game to study question generation, improving the quality of generated language using concepts related to latent action spaces. Some works like [27] and [28] also aim to ask visual questions with limited question supervision. Other works represent dialog using latent action spaces [16, 29, 30, 31, 32, 33, 31, 34, 35, 36, 37]. Finally, reference games are generally popular for studying language [38, 5, 25]. Section 6 of the supplement describes the relationship between our approach and these works in more detail.

## 6 Conclusion

In this paper we proposed the Dialog without Dialog (DwD) task along with a model designed to solve this task and an evaluation scheme that takes its goals into account. The task is to perform the image guessing game, maintaining language quality without dialog level supervision. This balance is hard to strike, but our proposed model manages to strike it. Our model approaches this task by representing dialogs with a discrete latent variable and carefully transfering language information via multi stage training. While baseline models either adapt well to new tasks or maintain language quality and intelligibility, our model is the only one to do both according to both automated metrics and human judgements. We hope these contributions help inspire useful dialog agents that can also interact with humans.

## 7 Acknowledgements

The Georgia Tech effort was supported in part by NSF, AFRL, DARPA, ONR YIPs, ARO PECASE, Amazon. The views and conclusions contained herein are those of the authors and should not be interpreted as necessarily representing the official policies or endorsements, either expressed or implied, of the U.S. Government, or any sponsor.

## 8 Broader Impact

We think the main ethical aspects of this work and their consequences for society have to do with fairness. There is an open research problem around existing deep learning models often reflecting and amplifying undesirable biases that exist in society.

While visual question answering and visual dialog models do not currently work well enough to be relied on in the real world (largely because of the aforementioned proneness to bias), they could be deployed in applications where these biases could have negative impacts on fairness in the future. For example, visually impaired users might use these models to understand visual aspects of their world [39]. If these models are not familiar with people in certain contexts (e.g., men shopping) or are only used to interacting with certain users (e.g., native English speakers) then they might fail for some sub-groups (e.g., non-native English speakers who go shopping with men) but not others.

Our research model may be prone to biases, though it was trained on the VQAv2 dataset [6], which aimed to be more balanced than its predecessor. However, by increasing the intelligibility of generated language our work may help increase the overall interpretability of models. This may help by making bias easier to measure and providing additional avenues for correcting it.

## Footnotes

*Equal contribution. Work carried out mainly at Georgia Tech.

[2]We find 5 epochs stops training early enough to avoid overfitting on our val set.

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
