[Supplementary Material]

# 9 Supplement Overview

This document contains supplementary material for "Dialog without Dialog Data: Learning Visual Dialog Agents from VQA Data". The main paper excludes some details which we provide here. Section 10 describes the Q-bot proposed in the main paper in more detail, including algorithms that show how it executes one round of dialog. Section 11 describes the human studies we use to evaluate our model and reports those results in detail. Section 12 reports the ablations we use to evaluate the effects of different aspects of the proposed Q-bot. Section 13 reports how various models we consider perform at different rounds of dialog. Finally, Section 14 explores in more depth how our work relates to other relevant work in the literature.

# 10 Architecture Details

This section describes our architecture in more detail. Algorithm 1 summarizes our complete `QBot()` implementation and subsequent algorithms define the subroutines used inside `QBot()` along with the encoder we use for variational pre-training. The planner module is described in Algorithm 3, the predictor is described in Algorithm 2, and the speaker is described in Algorithm 4. Algorithm 5 describes the encoder used for the ELBO loss.

Note that the number of bounding boxes per image is $B$, the number of images in a pool is $P$, and the max question length is $T$.

There is a minor notation difference between this section and the main paper. In this section there is an additional hidden state $\bar{h}_r$ that parallels $h_r$ and is used only inside the planner. While $h_r$ is the hidden state of an LSTM, $\bar{h}_r$ is computed in the same way except it uses a different output gate (see line 11 of Algorithm 3). This is essentially a second LSTM output that allows the context coder query to forget dialog history information irrelevant to the current round, and allowing $h_r$ to focus on representing the entire dialog state.

---

**Algorithm 1:** Question Bot

1 **Function** QBot $(\mathcal{I}, h_{r-1}, \bar{h}_{r-1}, q_0, a_0, \ldots, q_{r-1}, a_{r-1})$
    **Input:** $\mathcal{I}, h_{r-1}, \bar{h}_{r-1}, q_0, a_0, \ldots, q_{r-1}, a_{r-1}$
    **Output:** $q_r, h_r, \bar{h}_r, y_r$
2     $y_r \leftarrow$ Predictor $(\mathcal{I}, h_{r-1}, q_0, a_0, \ldots, q_{r-1}, a_{r-1})$
3     $h_r, \bar{h}_r, z^r \leftarrow$ Planner $(\mathcal{I}, q_{r-1}, a_{r-1}, h_{r-1}, \bar{h}_{r-1})$
4     $q_r \leftarrow$ Speaker $(z^r)$
5     **return** $q_r, h_r, \bar{h}_r, y_r$

---

**Algorithm 2:** Predictor

1 **Function** Predictor $(\mathcal{I}, h_{r-1}, q_0, a_0, \ldots, q_{r-1}, a_{r-1})$
    **Input:** $\mathcal{I}$ $(I_p^b \in \mathbb{R}^{2048})$, $h_{r-1}, q_0, a_0, \ldots, q_{r-1}, a_{r-1}$
    **Output:** $y_r$
2     $\text{Attention}(Q, K, V) = \text{Softmax}\left(g_3(g_1(Q) \odot g_2(K))\right) V$
3     $f_{r-1} \leftarrow [E_q(q_{r-1}), E_a(a_{r-1})]$ /* fact                             */
4     $F \leftarrow [f_0, \ldots, f_{r-1}]$
    /* Attention over rounds                                         */
5     $e_F \leftarrow \text{Attention}(h_{r-1}, F, F)$
6     $Q_y \leftarrow [h_{r-1}, e_F]$
    /* Attention over bounding boxes                           */
7     $e_I \leftarrow \text{Attention}(Q_y, \mathbf{x}, \mathbf{x}) \in \mathbb{R}^{P \times 2048}$
8     $e_I \leftarrow g_1(e_I)$
9     $Q_p \leftarrow g_2(Q_y)$
10     $l_y \leftarrow g_3(Q_p \odot e_I)$
11     $y_r \leftarrow \text{argmax Softmax}(l_y)$
12     **return** $y_r$

---

---
**Algorithm 3:** Planner
---

1 **Function** `Planner` $(\mathcal{I}, q_{r-1}, a_{r-1}, h_{r-1}, \bar{h}_{r-1})$

    **Input:** $\mathcal{I}$ $(I_b^p \in \mathbb{R}^{2048})$, $q_{r-1}, a_{r-1}, h_{r-1}, \bar{h}_{r-1}$

    **Output:** $h_r, \bar{h}_r, z^r$

    `/* Context Coder                                              */`

2    $e_q \leftarrow E_q(q_{r-1})$

3    $e_a \leftarrow E_a(a_{r-1})$

4    $e_c \leftarrow f_5([\bar{h}_{r-1}, e_q, e_a])$

5    $\alpha_p \leftarrow \mathrm{Softmax}\left(f_2(g(e_c) \odot f_1(I_b^p))\right)$

6    $\beta_b^p \leftarrow \mathrm{Softmax}\left(f_4(g(e_c) \odot f_3(I_b^p))\right)$

7    $\hat{v}_{r-1} \leftarrow \sum_{p=1}^{P} \sum_{b=1}^{B} \alpha_p \beta_b^p I_b^p$

8    $x_{r-1}^{context} \leftarrow [\hat{v}_{r-1}, e_q, e_a]$

    `/* Dialog RNN                                                  */`

9    $h_r, c_r \leftarrow \gamma(x_{r-1}^{context}, h_{r-1})$

10    $h_r \leftarrow \mathrm{Dropout}(h_r)$

11    $\bar{h}_r \leftarrow \sigma(W_1^T x_{r-1}^{context} + W_2^T h_{r-1}) \odot \tanh(c_r)$

12    $\bar{h}_r \leftarrow \mathrm{Dropout}(\bar{h}_r)$

    `/* Question Policy                                             */`

13    $l_{n,k} \leftarrow \mathrm{LogSoftmax}\,(W_n^z h_r)_k \ \forall k \in \{1, \ldots, K\} \ \forall n \in \{1, \ldots, N\}$

14    $z_n^r \leftarrow \mathrm{GumbelSoftmax}(l_n) \ \forall n \in \{1, \ldots, N\}$

15    $h_r \leftarrow h_r + \mathrm{ReLU}\left(W^l l\right)$

16    **return** $h_r, \bar{h}_r, z^r$

---

In the planner Algorithm 3 at lines 5 and 6 $g, f_1, f_3$ are all two layer MLPs with ReLU output and weight norm. Both $f_2$ and $f_4$ are linear transformations with weight norm applied (no activation function). $f_5$ is a linear transformation without weight norm purely for dimensionality reduction. To compute $\bar{h}_r$ we also add new linear weights $W_1$ and $W_2$ as for a standard LSTM output gate.

Note that for the planner there is an additional residual connection at line 16 which augments the hidden state. This allows gradients to flow through the question policy parameters $W^z$ at line 13 when we fine-tune for task performance without fully supervised dialogs.

In Algorithm 2 $g_1, g_2$ are both 2-layer ReLU nets with weight norm. Also $g_3$ is a 2-layer net with ReLU and Dropout on the hidden activation and weight norm on both layers.

---
**Algorithm 4:** Speaker
---

1 **Function** `Speaker` $(z)$

    **Input:** $z$

    **Output:** $q_{r+1}$

2    $e_z \leftarrow \sum_{n=0}^{N-1} E_n^z(z_n)$

3    $q_{r+1} \leftarrow \beta(e_z)$

4    **return** $q_{r+1}$

---

In Algorithm 4 $\beta$ is an LSTM decoder.

## 11 Human Evaluation

As summarized in Section 4.5 of the main paper, we also evaluate our models by asking if humans can understand Q-bot's language. Specifically, we use workers (turkers) on Amazon Mechanical Turk to evaluate the relevance, fluency, and task performance of our models. Fig. 5, Fig. 6 and Fig. 7 show the interface to evaluate question relevance, question fluency and interactive task performance respectively.

**Algorithm 5:** Encoder

```
1  Function Encoder (ℐ, q_r)
```

**Input:** $\mathcal{I}(I_p^b \in \mathbb{R}^{2048})$, $q_r$

**Output:** $z$ (sample or distribution parameters)

```
   /* Context Coder                                          */
```

$$2 \quad e_q \leftarrow E_q(q_r)$$

$$3 \quad \alpha_p \leftarrow \text{Softmax}\left(f_2(g(e_q) \odot f_1(I_b^p))\right)$$

$$4 \quad \beta_b^p \leftarrow \text{Softmax}\left(f_4(g(e_q) \odot f_3(I_b^p))\right)$$

$$5 \quad \hat{v} \leftarrow \sum_{p=1}^P \sum_{b=1}^B \alpha_p \beta_b^p I_b^p$$

$$6 \quad h \leftarrow W_z^T \hat{v}$$

$$7 \quad l_{n,k} \leftarrow \text{LogSoftmax}\left(W_n^z h\right)_k \ \forall k \in \{1,\dots,K\} \ \forall n \in \{1,\dots,N\}$$

$$8 \quad z_n \leftarrow \text{GumbelSoftmax}(l_n) \ \forall n \in \{1,\dots,N\}$$

```
9      return z
```

Figure 5: Human study instructions for question relevancy.

## 12  Model Ablations

We investigate the impact of our modelling choices from Section 3.2 of the main paper. In Tab. 2 we report the mean of all four automated metrics averaged over pool sizes, pool sampling strategies, and datasets.[3] Next we explain how we vary each of these model dimensions

Figure 6: Human study instructions for question fluency.

Figure 7: Interactive task performance MTurk interface.

– Our 128 4-way Concrete variables $z_1^r, \ldots, z_{128}^r$ require 512 logits (**Discrete**). Thus we compare to the standard Gaussian random variable common throughout VAEs with 512 dimensions (**Continuous**).

– In both discrete and continuous cases we train with an ELBO loss (**ELBO**), so we compare to a maximum likelihood only model (**MLE**) that uses an identity function as in the default option for the Question Policy (see Section 2.3.1 of the main paper). The MLE model essentially removes the KL term (2nd term of Eq. 7 of the main paper) and ignores the encoder during pre-training.

– We consider checkpoints after each step of our training curriculum: **Stage 1**, **Stage 2.A**, and **Stage 2.B**. For some approaches we skip Stage 2.A and go straight to fine-tuning everything except the speaker as in Stage 2.B. This is denoted by **Stage 2**.

|   | $z$ Structure | Loss | Curriculum | Speaker | Accuracy | Perplexity | Relevance | Diversity |
|---|---|---|---|---|---|---|---|---|
| 1 | Discrete | ELBO | Stage 2.B | Fixed | 0.81 | 2.57 | 0.89 | **0.86** |
| 2 | Discrete | ELBO | Stage 2 | Fine-tuned | **0.82** | 2.54 | 0.85 | 0.59 |
| 3 | Discrete | ELBO | Stage 2 | Parallel | 0.78 | 2.60 | 0.88 | 0.73 |
| 4 | Discrete | ELBO | Stage 1 | Fixed | 0.72 | 2.60 | **0.91** | 0.48 |
| 5 | Discrete | ELBO | Stage 2.A | Fixed | 0.80 | 2.59 | 0.89 | 0.81 |
| 6 | Discrete | ELBO | Stage 2 | Fixed | 0.80 | **2.53** | 0.85 | 0.62 |
| 7 | Continuous | ELBO | Stage 2.B | Fixed | 0.75 | 2.45 | 0.66 | 0.23 |
| 8 | Continuous | MLE | Stage 2.B | Fixed | 0.78 | 4.27 | 0.83 | 4.33 |

Table 2: Various ablations of our training curriculum.

– We consider 3 variations on how the speaker is fine-tuned. The first is our proposed approach of fixing the speaker (**Fixed**). The next fine-tunes the speaker (**Fine-tuned**). To evaluate the impact of fine-tuning we also consider a version of the speaker which can not learn to ask better questions by using a parallel version of the same model (**Parallel**). This last version will be described more below.

**Discrete Outperforms Continuous $z^r$.** By comparing our model in row 1 of Tab. 2 to row 7 we see that our discrete model outperforms the corresponding continuous model in terms of task performance (higher Accuracy) and about matches it in interpretability (similar Perplexity and higher Relevance). This may be a result of discreteness constraining the optimization problem to prevent overfitting and is consistent with previous work that used a discrete latent variable to model dialog [16].

**Stage 2.B Less Important than Stage 2.A** Comparing rows 4, 5, and 1 of Tab. 2, we can see that each additional step, Stage 2.A (row 4 –> 5) and Stage 2.B (row 5 –> 1), increases task performance and stays about the same in terms of interpretability. However, most gains in task performance happen between Stage 1 and Stage 2. This indicates that improvements in task performance are mainly from learning to incorporate information over multiple rounds of dialog.

**Better Predictions, Slightly Better Questions** To further investigate whether Q-bot is asking better questions or just understanding dialog context for prediction better we considered a **Parallel** speaker model. This model loaded two copies of Q-bot, A and B both starting at the model resulting from Stage 1. Copy A was fine-tuned for task performance, but every $z^r$ it generated was ignored and replaced with the $z^r$ generated by copy B, which was not updated at all. The result was that copy A of the model could not incorporate dialog context into its questions any better than the Stage 1 model, so all it could do was track the dialog better for prediction purposes. By comparing the performance of copy A (row 3 of Tab. 2) to our model (row 1) we can see a 3 point different in accuracy, so the question content of our model has improved after fine-tuning, but not by a lot. Again, this indicates most improvements are from dialog tracking for prediction (row 3 accuracy is much higher than row 4 accuracy).

**Fine-tuned Speaker** During both Stage 2.A and Stage 2.B we fix the Speaker module because it is intended to capture low level language details and we do not want it to change its understanding of English. Row 2 of Tab. 2 does not fix the Speaker during Stage 2 fine-tuning. Instead, it uses each softmax at each step of the LSTM decoder to parameterize one Concrete variable [17] per word. This allows gradients to flow through the decoder during fine-tuning, allowing the model to tune low-level signals. This is similar to previous approaches which either used this technique [40] or REINFORCE [2] This model is competitive with DwD in terms of task performance. However, when we inspect its output we see somewhat less understandable language.

**Variational Prior Helps Interpretability** We found the most important factor for maintaining interpretability to be the ELBO loss we applied during pre-training. Comparing the continuous Gaussian variable (row 7) to a similar hidden state (row 8) trained without the KL prior term we see drastically different perplexity and diversity. In the main paper these metrics dropped when a model had drifted from English (*e.g.*, for Typical Transfer). This suggests the model without the ELBO in row 8 has experienced similar language drift.

## 13 Performance by Round

Experiments in the main paper considered dialog performance after the first round (top of Table 1) and at the final round of dialog (either 5 or 9). This does not give much sense for how dialog

performance increases over rounds of dialog, so we report `QBot()`'s guessing game performance at each round of dialog in Fig. 8. For all fine-tuned models performance goes up over multiple rounds of dialog, though some models benefit more than others. Stage 1 models decrease in performance after round 1 because it is too far from the training data such models have been exposed to.

Figure 8: Task performance (guessing game accuracy) over rounds of dialog. Performance increases over rounds for all models except the Stage 1 models.

## 14 Related Work

We used a visual reference game to study question generation, improving the quality of generated language using concepts related to latent action spaces. This interest is mainly inspired by problems encountered when using models comparable to the Zero-shot Transfer baseline. In [3] a dataset is collected with question supervision then fine-tuning is used in an attempt to increase task performance, but the resulting utterances are not intelligible. Similarly, [2] takes a very careful approach to fine-tuning for task performance but finds that language also diverges, becoming difficult for humans to understand.

**Visual Question Generation.** Other approaches like [27] and [28] also aim to ask questions with limited question supervision. They give Q-bot access to an oracle to which it can ask any question and get a good answer back. This feedback allows these models to ask questions that are more useful

for teaching A-bot [27] or generating scene graphs [28], but they require a domain specific oracle and do not take any measures to encourage interpretability. We are also interested in generalizing with limited supervision, using a standard VQAv2 [6] trained A-bot as a flawed oracle, but we focus on maintaining interpretability of generated questions and not just their usefulness.

**Latent Action Spaces.** Of particular interest to us is a line of work that uses represents dialogs using latent action spaces [29, 30, 31, 32, 33, 31, 34, 35, 36, 37]. Recent work uses these representations to discover intelligible language [29] and to perform zero-shot dialog generation [30], though neither works consider visually grounded language as in our approach. Most relevant is [16], which focuses on the difference between word level feedback and latent action level feedback. Like us, they use a variationally constrained latent action space (like our $z$) to generate dialogs and find that by providing feedback to the latent actions instead of the generated words (as opposed to the approaches in [2] and [3]) they achieve better dialog performance. Our variational prior is similar to the Full ELBO considered in [16], but we consider generalization from non-dialog data and generalization to new modalities.

**Reference Games.** The task we use to study question generation follows a body of work that uses reference games to study language and its interaction with other modalities [38]. Our particular task is most similar to those in [5] and [25]. In particular, [5] collects a dataset for goal oriented visual dialog using a similar image reference game and [25] uses a similar guessing game we use to evaluate how well humans can interact with A-bot.

## Footnotes

[3] This includes 10 settings: {random 2, 4, 9 pools } × {VQA, AWA, CUB} and 2 contrast pools on VQA