[Reviews · NeurIPS 2020]

Review 1

Summary and Contributions: Method for training a visual dialog system with only VQA data and not dialog data. Experiments confirm effectiveness of approach.

Strengths: The idea of the DwD task in interesting, novel, and compelling. The novel architecture encourages appropriate generalization from VQA training to dialog adaptation without dialog training data. Experiments are quite comprehensive, including a variety of automated metrics and human evaluation and ablations of the method.

Weaknesses: The model seems quite complicated, maybe overly so, but the ablation experiments illustrate the importance of several aspects of this complex model. Why use a 2-year old bottom-up-top-down VQA model? This seems very dated now. A more recent multimodal transformer model like Vilbert, LexMert, or Uniter would be more appropriate. Wouldn't a more modern model help produce better results overall?

Correctness: Paper seems technically sound and experimental methodology is reasonable including a variety of automated and human eval metrics and ablations. My biggest question about the paper is: Why are there no experiments directly evaluating the Q-bot's ability to engage an actual human user in a complete dialog, demonstrating its ability to engage a user in a full dialog, evaluating it's ability to find a hidden image given to the user. This sort of full user-study on the end-to-end task would be the most compelling evaluation and would greatly improve the quality of the paper. It is not clear why it is missing.

Clarity: The paper is fairly clear and well written, but it tries to cram a lot of information in a short paper leaving many details for the appendix. I feel a paper that focuses the evaluation more on a final human user study as outlined above would be better, leaving some of the other experiments for the appendix.

Relation to Prior Work: Good coverage of related work.

Reproducibility: Yes

Additional Feedback:


Review 2

Summary and Contributions: The paper addresses the problem of learning one type of task and adapting it to another task. Specifically, the paper focuses on learning to perform visual dialogue without having learned to do the dialogue. Using VQA data, it learns to ask discriminative questions. Then question intent is used to generated new questions in the dialogue. The evaluation is performed both using automatic metrics for both task success and language drift. Also, human evaluation is performed to check the language drift.

Strengths: + paper is written well and understandable + Using VQA data to pre-train the dialogue model is an interesting approach + Having "Not Relevant" answer in A-Bot could help Q-Bot's planner to filter out asking the non-relevant question

Weaknesses: + The main problem with the paper is the game design. In visual dialogue, i.e GuessWhich game[2], does not have access to the image. It has to build up the visual representation based on the caption and dialogue. That is why having a caption is important for the GuessWhich game (L69). While in the proposed game, since Q-Bot has constant access to the images. It just needs to ask questions such that it distinguished the one image from the other. Which mean that ask questions such that narrow down the type of image (bird from car) and then try to differentiate among different similar type image (say bird). This could be observed from the questions generated from Figure 3. + Since the model is trained on VQA data to distinguish images, it only learns to ask the discriminative type of questions. However, the dialogue is not always discriminative; for example, the dialogue could also involve clarification, follow-up question etc. That means that model is not learning these types of skills only focusing on one type of skill. + The way distractor images are selected is not systematic. Because of random selection, some of the games might be very easy to perform, only need to distinguish b/w contrasting VQA images. For VQA (i.e., MS-COCO) images, object category, super-categories, etc, could be used to create systematic and challenging distractor images. Look at "Multimodal Hierarchical Reinforcement Learning Policy for Task-Oriented Visual Dialog" for some suggestions. + As we could see from Table 1, B3 vs. C3. Having random distractor significantly improves performance. So having challenging distractors will have a greater effect on the performance. + Dipper analysis on the distractor, for eg in the case of CUB does having bird image as target images make tasks more challenging. + Results are not clearly explained in the text. For example, L266 "Longer dialogs (more rounds) achieve better accuracy". However, it is not supported clearly. It would have been better to see results for 9 distractors with a different number of rounds of dialogue and all measures how it performs say as a graph. + Make it clear how the target image is selected for A-Bot (L62-63) + Missing dataset splits is paper using the same split as GuessWhich. Please provide details. + For language analysis looks at "The Devil is in the Details: A Magnifying Glass for the GuessWhich Visual Dialogue Game," which measures Lexical Diversity (similar to language-diversity), Question diversity etc. + Table 1 caption "Our method strikes a balance between guessing game performance and interpretability." Please clarify how interpretability is improved. + Fig 3 provide Q4 answer and guessed image by models

Correctness: Looks okay.

Clarity: Well written paper. Need some improvement in result presentation and explanation.

Relation to Prior Work: Yes

Reproducibility: Yes

Additional Feedback:


Review 3

Summary and Contributions: This paper presents a factorized training approach to training a question-generating agent for a GuessWhich? style image guessing game that first pretrains on single question-answer pairs and then fine-tunes on visual dialog data. The proposed approach maintains higher language fluency and relevance than standard fine tuning (which exhibits language drift towards agent-agent 'neuralese') while achieving higher accuracy than zero shot transfer (which has high fluency but does not know how to sequence questions for dialog). Interestingly, the method also outperforms typical transfer on task accuracy in most tested settings across two datasets and various game and dialog parameter settings. **Post-rebuttal: the rebuttal addresses questions and concerns from other reviewers and suggests the additional space on acceptance can be partially used to round out details currently relegated to the supp. I continue to feel the paper should be accepted.

Strengths: The use of human evaluations to decisively measure language intelligibility is a major strength of the paper's evaluation. It would be great to see it elevated into a row of a results table.

Weaknesses: The paper leans heavily on prior work in VQA, GuessWhich?, etc., and so many implementation details are referred to as "standard" or detailed only in the supplement. This may limit the accessibility of the contribution, and limits its scope without the inclusion of a broader discussion of where this kind of approach could succeed outside of the small game setting presented.

Correctness: Yes, the evaluation is thorough.

Clarity: Yes, the paper is easy to follow and the contribution is clear.

Relation to Prior Work: Yes, the paper situates itself well with respect to existing work in VQA and image selection games.

Reproducibility: Yes

Additional Feedback: For adding a "not relevant" prediction, from my understanding, a single randomly sampled question per image is added to the training set with "not relevant" as an answer. Doesn't this add a weird latent prior on the frequency of "not relevant" that's a function of the training questions available on average per training image? Anyway, it's neat that this seemed to just sort of work out of the box like that. It's interesting that this method tends to have higher accuracy (Table 1) versus typical transfer. Is the transfer baseline crippled somehow or is this just explained by better generalization to the test data? Seeing final training numbers here might help tease that out, either in the main Table or in the supplement with a small explanation. Line 337 is making a good point in a really hand-wavy fashion. Being more concrete here would be helpful, since the way it's written sort of excuses models and doesn't explain why these arise (e.g., sample bias versus color space representation bias). nits: Line 3 missing word "a" Figure 3 misalignment in row 1 box "Ours" on final A3/P3 Line 271 lowercased "a-bot" Line 273 n-dash should be a comma or the word "and" Citations include both arXiv-style and CoRR style for arXiv papers. Some citations are listed as on arXiv only when they have been published (e.g., [12] appeared at ACL'20, which happened after this article was submitted, so references need to be updated accordingly; [29] appeared at ICML'18 but that isn't reflected here; etc.).


Review 4

Summary and Contributions: This papers tackles the problem of language drift when optimizing chat bots in task-oriented dialogues based on a task completion scores. The proposed approach decouples text generation from policy learning. Specifically, the policy is a variational auto-encoder with discrete states. The text generator converts discrete states to natural language; it's pre-trained and fixed during policy learning to ensure that the language does not drift away from natural language. The approach is tested on an image guessing game. ========================== Thanks for the response and comment on VAE with discrete spaces.

Strengths: This work tackles a challenging problem in RL-based text generation. Typically, when we optimize the reward, the objective does not necessarily limit the generation to natural language, which may produce a model that speaks in "bot language". I like the proposed approach in that it decouples text generation from policy learning in an elegant way through discrete policy states and the pre-trained text generator, such that the text generation part can be left untouched during policy learning.

Weaknesses: I think the proposed approach might have two potential problems in practice and I would like to see more discussion on how it is solved in the image guessing setting. First, VAEs often has the mode collapse problem where the latent structure is ignored. Did it happen here? If not, do the learned states have any task-specific meaning? Second, during policy learning the latent states are likely to deviate from their pre-training distribution, which may cause problems for the text generator (which is fixed during policy learning). Is the text generator robust to distribution shift of the latent states?

Correctness: Yes, the approach makes sense to me, although I'd like to see more explanation on potential pitfalls in practice.

Clarity: The paper is quite clear and easy to read.

Relation to Prior Work: Yes.

Reproducibility: No

Additional Feedback: I'm surprised to see that Typical Transfer has low accuracy in quite a few settings. They are prone to language drift, but should be able to optimize the reward, right? I think an important baseline is also missing. One standard way to alleviate language drift in policy learning is to interpolate that with MLE training. So here Typical Transfer could be interpolated with MLE training on VQA (same dataset used in pre-training). There will be a trade-off between optimizing the reward and avoiding language drift though. Still, this is commonly used in practice and should be compared against.

[Author Response · NeurIPS 2020]

We thank the reviewers for the thoughtful feedback! We are encouraged that all voted to accept, finding the DwD task interesting [**R1**,**R2**], novel, and compelling [**R1**,**R4**]; our approach elegant [**R4**] and interesting [**R2**]; and our experiments comprehensive [**R1**]. [**R3**] appreciates the use of human evaluations to decisively measure language interpretability. We respond to select comments below but will address all feedback.

**[R1] Why not use a more recent VQA model? Performance would improve.** Yes, it likely would; however, this is orthogonal to our primary investigation. The focus of our work is on adapting Q-bot's questioning strategy to a dialog without having seen dialog during training. The BUTD model is a well-established model to demonstrate this on. We agree doing so with more recent transformer-based models is an interesting future direction.

**[R1] Why no human experiments evaluating human & Q-bot pairs on game performance?** These experiments were reported in Sec 4.4. We paired humans with Q-bots and ran the game with humans responding to Q-bot's questions as the reviewer describes. As in L284, game performance with a human answering Q-bot's questions is 69% for our method, 45% for typical transfer, and 23% for zero-shot transfer. This result is a key finding of this work and highlighted in Contribution 3 in the introduction. We also had the human players evaluate the fluency and relevance of questions.

**[R2] Game design includes access to the image pool for the questioner unlike in visual dialog.** We see this as a strength not a weakness. A question may be discriminative in one pool, but not in another, so questions should depend on the pool. By adopting a pool-conditioned setting, we can evaluate Q-bot's adaptation to different pool sizes, image domains, and pool selection strategies. In contrast, pool-free methods in prior work will always produce the same question for an input regardless of the pool once trained – implicitly conditioning on the training pool. We also note that past pool-free work has found that access to the caption results in the subsequent dialog not playing a significant role.

**[R2] The model is trained on VQA data that only focused on discriminative questioning, not other aspects of dialog.** Exploring other aspects of dialog (e.g., clarification questions) is interesting future work. That said, our model does exhibit continuity in the dialog. For instance, we visualize the dialog and find our Q-bot asks "is the woman alone?" followed by "what is she holding in her hand ?". The hand refers to the hand of the woman from the earlier question.

**[R2] Random distractor images may be easier than a more systematic selection.** We agree, but harder pools would make the task harder for all approaches. Our focus is on the relative performance of methods rather than the absolute performance. Note that we tried a harder pool by selecting visually similar images and the models' trends are similar, though overall performance was worse.

**[R2] "Longer dialogs achieve better accuracy" is not supported clearly.** Due to page limit, we show the task performance over rounds of dialog in Figure 8 of supplement. Performance generally goes up for Stage 2 models (trained for multiple rounds), but it goes down for Stage 1 models (only trained for a single round). This trend is very consistent across different models.

**[R2] Make it clear how the target image is selected for A-Bot.** Will do. The target image is randomly selected.

**[R2] Missing dataset splits. Is paper using the same split as GuessWhich?** Since our task contains out-of-domain images such as AWA and CUB, we can not use the same split as GuessWhich. For COCO, we use default train split and randomly split the val split into validation (30%) and test (70%). For AWA and CUB, we use the same train/val/test splits defined by the datasets.

**[R2] Please clarify how interpretability is improved.** We measured interpretability using fluency metrics and human performance / qualitatives (Section 4.4). Our model is able to generate questions which are more relevant to the image pool and more fluent compared to the Typical Transfer model. Furthermore, when humans are paired with our model they perform better at the game than when paired with a baseline models. This directly demonstrates interpretability: humans are able to interpret our model's responses better.

**[R3] Many implementation details are referred to as "standard" or detailed only in the supplement. This may limit the accessibility of the contribution.** Thanks for this valuable perspective. If granted the additional page provided to accepted papers, we will try to make more of this background available to readers.

**[R4] VAEs often have the mode collapse problem where the latent structure is ignored. Did it happen here?** We did not observe this in our experiments. Further, we find the latent space to be fairly well-behaved when interpolating. For example, interpolating in the latent space from "how many beds?" to "where is he looking?" yields this result after removing the replicated questions: *how many beds? - how many cats? - how many dogs? - where is the dog? - where is the man? - where is this man? - where is this woman? - where is this? - where is he? - where is he looking?*

**[R4] Is the text generator robust to the distribution shift of the latent states?** Based on the interpolation result above, we believe so. We hypothesize this is because the discrete latent representation and VAE pre-training help disentangle intent from expression by restricting information flow through $z$.

**[R4] Typical Transfer has low accuracy in quite a few settings.** Our hypothesis is that what enables Typical Transfer to achieve high performance for VQA is its ability to find patterns that "overfit" to Abot. These overfitting patterns are harder to find when the domain shifts (AWA/CUB).

[Meta-Review · NeurIPS 2020]

All reviewers agree that this submission is above the acceptance threshold and they are all agree that the idea of decoupling text generation from policy learning during RL is a compelling idea and interesting idea. I would also like to recommend acceptance with two notes: 1) the reviewers raised a number of questions which were addressed in the author response, most of which are already contained in the Supplementary material, so I would advice the authors to incorporate these points in the main manuscript 2) I see your method as a way to also deal with language drift more generally. There are a couple of recent papers looking into dealing with language drift. For example, Lee et al (2019) deal with language drift through image grounding while Lazaridou et al (2020) and Lu et al. (2020) also decouple generation and policy learning, the former through reranking of language modelling samples using the RL reward and the latter through distillation such that the RL signal is never disrupting the core language knowledge. Are any of these methods superior over the others? We don't know but it is a perhaps an interesting question for this paper to put out there. Lee, Jason, Kyunghyun Cho, and Douwe Kiela. "Countering language drift via visual grounding." arXiv preprint arXiv:1909.04499 (2019). Lazaridou, Angeliki, Anna Potapenko, and Olivier Tieleman. "Multi-agent Communication meets Natural Language: Synergies between Functional and Structural Language Learning." arXiv preprint arXiv:2005.07064 (2020). Lu, Yuchen, et al. "Countering language drift with seeded iterated learning." arXiv preprint arXiv:2003.12694 (2020).